# Coronary artery established through amniote evolution

Kaoru Mizukami[1], Hiroki Higashiyama[1]*, Yuichiro Arima[1,2], Koji Ando[3], Norihiro Okada[4], Katsumi Kose[5], Shigehito Yamada[6], Jun K Takeuchi[7], Kazuko Koshiba-Takeuchi[8], Shigetomo Fukuhara[3], Sachiko Miyagawa-Tomita[1,9,10], Hiroki Kurihara[1]*

[1]Department of Physiological Chemistry and Metabolism, Graduate School of Medicine, The University of Tokyo, Tokyo, Japan; [2]Developmental Cardiology Laboratory, International Research Center for Medical Science, Kumamoto University, Kumamoto, Japan; [3]Department of Molecular Pathophysiology, Institute of Advanced Medical Sciences, Nippon Medical School, Tokyo, Japan; [4]School of Pharmacy, Kitasato University, Tokyo, Japan; [5]Institute of Applied Physics, University of Tsukuba, Tsukuba, Japan; [6]Congenital Anomaly Research Center, Kyoto University Graduate School of Medicine, Kyoto, Japan; [7]Molecular Craniofacial Embryology, Graduate School of Medical and Dental Sciences, Tokyo Medical and Dental University, Tokyo, Japan; [8]Faculty of Life Sciences, Department of Applied Biosciences, Toyo University, Gunma, Japan; [9]Heart Center, Department of Pediatric Cardiology, Tokyo Women's Medical University, Tokyo, Japan; [10]Department of Animal Nursing Science, Yamazaki University of Animal Health Technology, Tokyo, Japan

*For correspondence:
h-hiroki@m.u-tokyo.ac.jp (HH);
kuri-tky@umin.net (HK)

Competing interest: The authors declare that no competing interests exist.

**Abstract** Coronary arteries are a critical part of the vascular system and provide nourishment to the heart. In humans, even minor defects in coronary arteries can be lethal, emphasizing their importance for survival. However, some teleosts survive without coronary arteries, suggesting that there may have been some evolutionary changes in the morphology and function of coronary arteries in the tetrapod lineage. Here, we propose that the true ventricular coronary arteries were newly established during amniote evolution through remodeling of the ancestral coronary vasculature. In mouse (*Mus musculus*) and Japanese quail (*Coturnix japonica*) embryos, the coronary arteries unique to amniotes are established by the reconstitution of transient vascular plexuses: aortic subepicardial vessels (ASVs) in the outflow tract and the primitive coronary plexus on the ventricle. In contrast, amphibians (*Hyla japonica*, *Lithobates catesbeianus*, *Xenopus laevis*, and *Cynops pyrrhogaster*) retain the ASV-like vasculature as truncal coronary arteries throughout their lives and have no primitive coronary plexus. The anatomy and development of zebrafish (*Danio rerio*) and chondrichthyans suggest that their hypobranchial arteries are ASV-like structures serving as the root of the coronary vasculature throughout their lives. Thus, the ventricular coronary artery of adult amniotes is a novel structure that has acquired a new remodeling process, while the ASVs, which occur transiently during embryonic development, are remnants of the ancestral coronary vessels. This evolutionary change may be related to the modification of branchial arteries, indicating considerable morphological changes underlying the physiological transition during amniote evolution.

## Editor's evaluation

Mizukami et al. investigate the evolutionary origins of coronary arteries in amniotes by comparing vascular morphologies across several species and developmental timepoints. They propose that coronary arteries are a novel structure in amniotes and that they arose from an ancestral vascular

network surrounding the outflow tract. The ancestral network is similar to, and may be a remnant of, the transient aortic subepicardial vessels (ASVs) seen in developing mouse and quail hearts.

## Introduction

Coronary circulation refers to the network of blood vessels that supply oxygen and nutrients to the heart muscle. In humans, disturbances in coronary flow caused by atherosclerosis and associated thrombosis lead to coronary artery disease, which can be lethal (*Nabel and Braunwald, 2012*; *Khera and Kathiresan, 2017*), indicating that coronary arteries distributed on the ventricles are essential for our survival. Despite their importance, the evolutionary origin of the coronary arteries remains uncertain.

Because of the complete absence of coronary circulation in cyclostomes (lampreys and hagfishes), the coronary artery is generally considered a synapomorphy of jawed vertebrates (*Grant and Regnier, 1926*). However, there is an inconsistency between amniotes and non-amniotes. In humans, the branching points (orifices) are located at the aortic sinuses close to the ventricle, and the arteries are distributed on the ventricular surface. Although the branching patterns of the vessels are diverse, the rostro-caudal level of the orifice and the distribution of the arteries on the ventricles are conserved in mammals, birds, crocodiles, and lepidosaurians (*Figure 1A*; *Spalteholz, 1924*; *Erhart, 1935*; *MacKinnon and Heatwole, 1981*; *Farrell et al., 2012*). In teleosts and chondrichthyans, however, the orifices are located more cranially, with branching from the hypobranchial arteries, and the arteries pass long distances from the orifices to the heart (*Parker, 1886*; *Grant and Regnier, 1926*; *Corrington, 1930*; *May and Herber, 1957*; *Halpern and May, 1958*; *De Andrés et al., 1990*; *Muñoz-Chápuli et al., 1994*; *Icardo, 2017*). Such positional differences have led to two hypotheses: coronary arteries gradually shifted from the rostral to the caudal position (*Halpern and May, 1958*; *MacKinnon and Heatwole, 1981*), or non-homologous arteries nourishing the heart are collectively termed coronary arteries (*Grant and Regnier, 1926*; *May and Herber, 1957*). Data on amphibians may help to address this controversy, although few comparative studies have examined their extrinsic blood vessels, and some authors have mentioned that amphibians lack coronary arteries (*Kapuria et al., 2018*; *Lupu et al., 2020*).

Coronary arteries are morphologically diverse and are frequently lost among teleosts, indicating that they are not essential for survival in this group, unlike in amniotes (e.g. *Grant and Regnier, 1926*). Thus, the knowledge of the morphological evolution of coronary arteries may contribute to our understanding of physiological changes in the heart during the water-to-land transition. Here, we compared the anatomy and development of the coronary vasculature in various vertebrate lineages to clarify the evolution of the unique coronary circulation of amniotes.

## Results

### Early development of cardiac vessels is comparable in mice and frogs

We first compared the anatomy of mice (*Mus musculus*) and Japanese quails (*Coturnix japonica*) to confirm the morphology of the coronary arteries. In a 17.5 days post-coitum (dpc) murine fetus, the two coronary arteries branched from the root of the aorta, part of the outflow tract (*Figure 1B* and *Figure 1—figure supplement 1*). They were distributed on the surface of the ventricle. Similarly, in quail at stage 28, the two coronary arteries branched from the aortic root and were distributed on the ventricular wall (*Figure 1B*). Their branching points and distribution patterns were identical to those in lizards and snakes (*Figure 1A*; see *MacKinnon and Heatwole, 1981*).

To investigate the early development of coronary arteries in mice, we performed immunohistochemical staining of pharyngula-stage embryos with antibody against PECAM-1. We found two distinct vessel structures: aortic subepicardial vessels (ASVs) and the primitive coronary plexus (The reconstructed image is in *Figure 2A* also see *Figures 2B and 3A, B*; *Figure 2—figure supplement 1*), in agreement with *Chen et al., 2014*. By 10.5 dpc, the primitive coronary plexus was already visible as a dotted pattern on the ventricular surface. The ASVs formed plexuses surrounding the pharyngeal arteries and one small vessel opening to the outflow tract at the root of the left branch of the truncus arteriosus (arrowheads in *Figure 2A*, *Figure 2—figure supplement 1A*). At 11.5 dpc, the position of the ASV orifice was identical to that at 10.5 dpc (arrowheads). No vasculature was detected on the

**eLife digest** Coronary arteries are tasked with supplying the heart with oxygenated blood and nutrients. Any blockage or developmental problem in these blood vessels can have severe and some-times lethal consequences. Due to their importance for health, researchers have extensively studied how coronary arteries form in humans and mice; a more limited range of studies have also looked at their equivalent in zebrafish. However, little is known about these structures develop in animals such as birds, amphibians, or other groups of fish. This makes it difficult to retrace the evolutionary processes that have given rise to the coronary arteries we are familiar with in mammals.

To address this knowledge gap, Mizukami et al. set out to compare blood vessel development around the heart of mammals, birds, amphibians, and fish. To do this, they performed detailed anatomical studies of blood vessel structure at different stages of development in mice as well as quail, frogs and newts, zebrafish and sharks.

In both mice and quail, small arterial subepicardial vessels (or ASVs) emerged early in development around the heart; these subsequently reorganised and remodelled themselves to give rise to the 'true' coronary arteries characteristic of the mature heart.

Frogs and newts also developed similar ASV-like structures; however, unlike their mammalian and bird equivalents, these vessels did not reorganise, instead being retained into adulthood. In fish, blood vessel development resembled that of amphibians, suggesting that the coronary artery-like structures seen in some fish are an 'ancestral' form of ASVs, rather than the equivalent of the mature coronary arteries in mammals and birds.

This work sheds light on the evolutionary processes shaping essential structures in the heart. In the future, Mizukami et al. hope that this knowledge will help develop a greater range of experimental animal models for studying heart disease and potential treatments.

outflow trunk at 10.5 and 11.5 dpc, indicating that the ASVs of the pharyngeal arches and ventricular primitive coronary plexus were physically separated (*Figure 2A*, *Figure 2—figure supplement 1A*). By 12.5 dpc, the ASVs extended from the orifice on the outflow trunk (aorta), and their peripheral end was anastomosed with the primitive coronary vessels (arrow in *Figure 2A*).

We also examined the development of the cardiac vasculature in Japanese tree frogs (*Hyla japonica*). In this species, larvae (tadpoles) are aquatic, but adults are terrestrial. In tadpoles (stage 33), we observed a small vasculature at the point of outflow tract bifurcation, with an orifice at the root of the left branch of the truncus arteriosus (*Figure 2Ca*). During metamorphosis (stage 42) and at the terrestrial juvenile stage (stage 46), the morphology of the truncus arteriosus did not change much, and the position of the orifice remained unchanged. At stage 46, the vessels that extended from the orifice were distributed on the outflow tract as truncal coronary vessels. On the basis of the positions of the orifices and the distribution patterns, these outflow truncal vessels in frogs were comparable to ASVs in mice. In contrast, no primitive coronary plexus was observed throughout the development in frogs.

## Coronary arteries are established in amniotes by remodeling of ASVs and primitive coronary vessels

To understand late development of the coronary vasculature in amniotes, we conducted whole-mount immunohistochemistry with PECAM-1 staining in mice and quails. In mice at 13.5 dpc, the proximal part of the ASVs degenerated and the cranial half of the aorta became avascular (*Figure 3A*). Subsequently, ASVs disappeared and ventricular coronary arteries were established (*Figure 1—figure supplement 1*). The development of quail was consistent with that of mice (*Figure 3B*). At stage 19, no QH-1-positive cells were observed on the surface of the outflow tract and a small vascular network was found around the pharyngeal arches. At stage 27, QH-1-positive endothelial cells formed a fine vascular plexus from the cranial part of the outflow tract toward the aortic root. At stage 35, the arterial plexuses of the aorta and pulmonary trunk degenerated and coronary arteries were formed secondarily on the ventricular surface.

The above developmental process was consistent with that reported in previous studies in mice. Namely, a novel orifice is formed by an ingrowth from the connection of the ASVs and the primitive

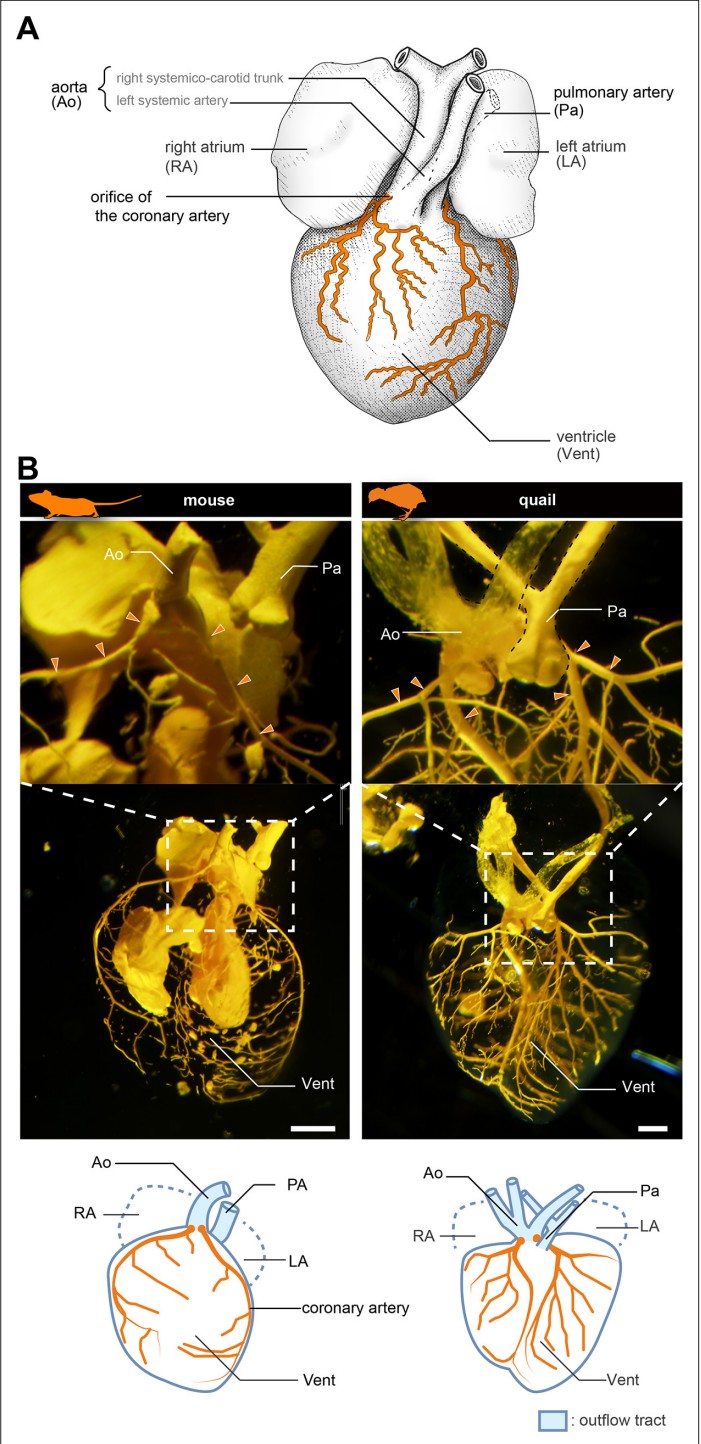

**Figure 1.** Anatomy of coronary arteries of amniotes. (**A**) Scheme of lizard coronary artery based on the data for the lace monitor (*Varanus varius*). In this lizard, a single coronary artery originates from the root of the right systemicocarotid trunk (the aorta in mammals) close to the heart and bifurcates almost at its point of origin to form a ventral and a dorsal division. The figure is based on *MacKinnon and Heatwole, 1981*, with some modifications for clarity: we drew shadings and added the position of the pulmonary artery. (**B**) Resin-injected coronary vessels of the mouse (17.5 dpc) and quail (stage 28) embryos. Scale bars: 1 mm.

The online version of this article includes the following figure supplement(s) for figure 1:

**Figure supplement 1.** Development of murine coronary artery visualized in latex-injected fetuses.

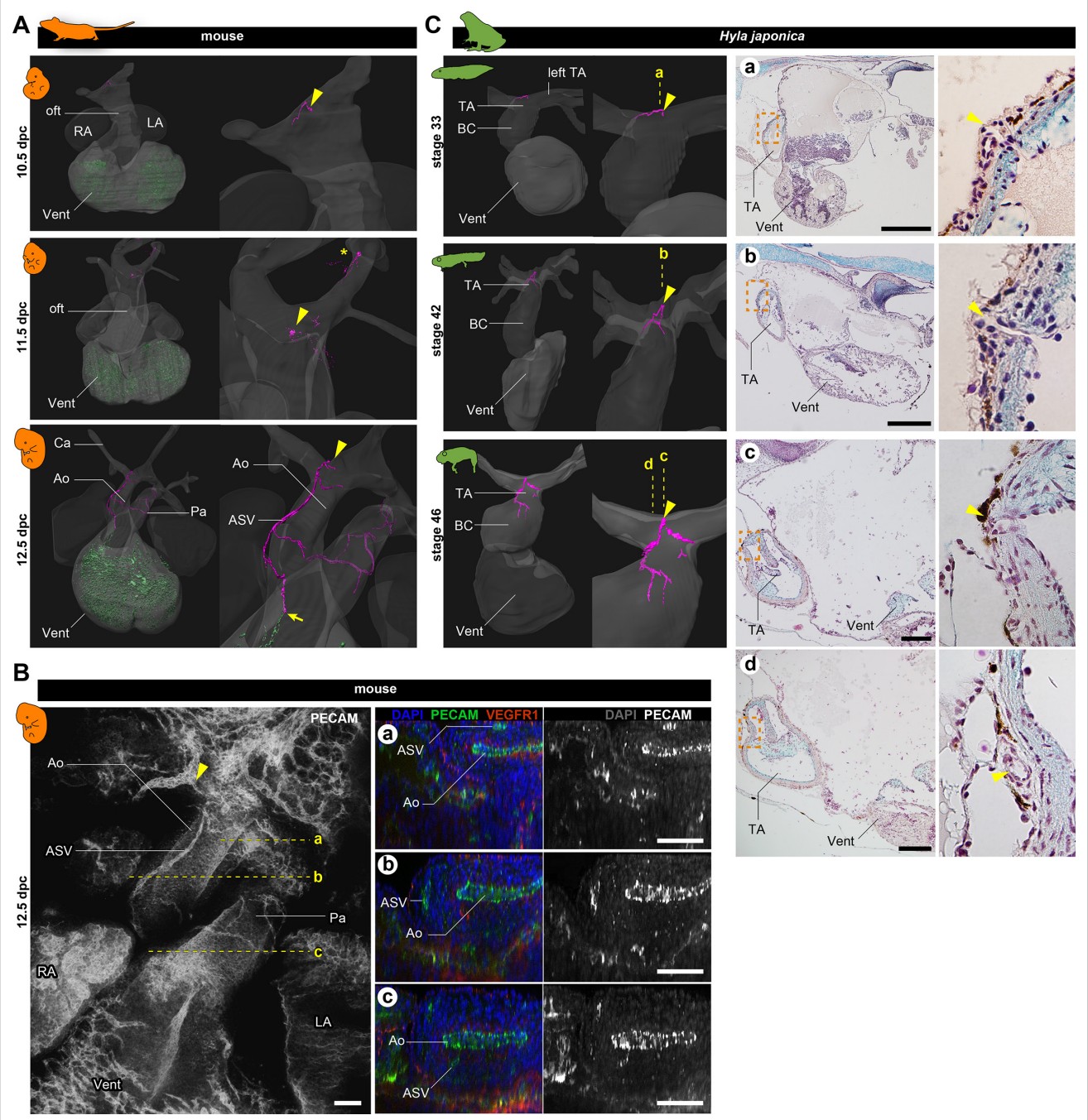

**Figure 2.** Development of vasculature in the outflow tract of mice and frogs. (**A**) Ventral views of three-dimensional reconstructed images of murine hearts. We visualized two types of primitive blood vessels: aortic subepicardial vessels (ASVs; pink) and primitive coronary plexuses (green). Although a plexus of vessels surrounded the ASVs, we visualized only the thickest vessels connected to the orifice (pink, indicated with arrowheads) to facilitate recognition. The ASV orifices were located at the bifurcation point of the outflow tract. We also found an extra orifice in the left carotid artery at 11.5 dpc (asterisk). At 12.5 dpc, the peripheral end of the ASV formed an anastomosis with the primitive coronary plexus at the outflow tract–ventricle boundary (arrow). Blood vessels were visualized using immunohistochemical staining for PECAM-1 (panel B and *Figure 2—figure supplement 1*). (**B**) Fluorescence images used to construct the three-dimensional images in (**A**). Transverse sections at the levels a–c in the left panel are shown in the right panels. The peripheral end of the ASV merged into the aorta (**c**). (**C**) Ventral views of three-dimensional reconstructed images of the hearts of the Japanese tree frog (*Hyla japonica*) constructed from histological sections show small vessels located on the outflow tract with orifices at the same location as in mice (arrowheads). No primitive coronary plexuses were found. Ao, aorta; BC, bulbus cordis; Ca, carotid artery; LA, left atrium; oft, outflow tract; Pa, pulmonary artery; RA, right atrium; TA, truncus arteriosus; Vent, ventricle. Scale bars: 500 µm.

The online version of this article includes the following figure supplement(s) for figure 2:

**Figure supplement 1.** Whole-mount immunohistochemistry with PECAM-1 in mouse embryos.

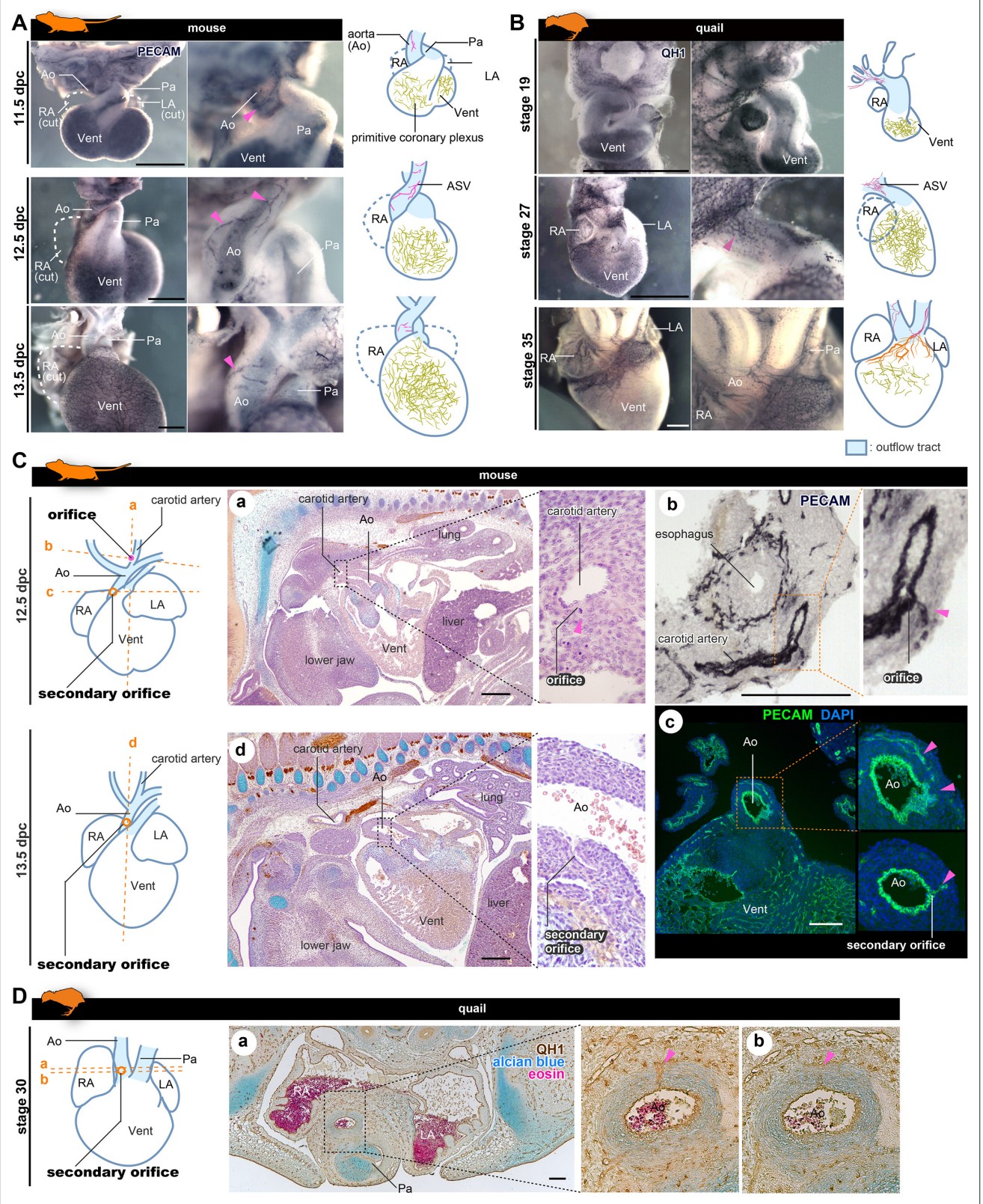

**Figure 3.** The positions of the orifices of the coronary arteries. (**A, B**) ASVs and primitive coronary plexuses are present in the embryonic stages in mice (**A**) and quails (**B**). The blood vessels were visualized by immunohistochemistry. Pink arrowheads indicate ASVs originated from the cranial part of the outflow tracts (same to the following panels). (**C**) ASVs in 12.5 dpc mouse embryos branched from the root of the future carotid artery (**a**). Immunohistochemical staining of cryosections shows two orifices: at the root of the carotid artery (**b**) and at the boundary of the outflow tract and

*Figure 3 continued on next page*

*Figure 3 continued*

ventricle (**c**). In 13.5 dpc embryos, the secondary orifice developed at the aortic root instead of the carotid artery orifice, which was lost (**d**). (**D**) In quails at stage 30, the secondary orifice was formed at the aortic root, as in mice. Immunohistochemical and HE staining images of cryosections (**a and b**). Ao, aorta; LA, left atrium; RA, right atrium; Vent, ventricle. Scale bars: 500 μm (**A, B**), 100 μm (**C, D**).

coronary plexus, and remodeling results in the secondary formation of the adult ventricular coronary arteries (*Bogers et al., 1989*; *Red-Horse et al., 2010*; *Peng et al., 2013*; *Tian et al., 2013*; *Chen et al., 2014*; *Ivins et al., 2015*; *He and Zhou, 2018*). To confirm this pattern and to understand whether it is common to other amniotes, such as birds, we conducted histological analysis in mice and quails. In 12.5 dpc mouse embryos, the ASV orifice was found at the root of the carotid artery (*Figure 3Ca*) and contained PECAM-1-positive endothelial cells (*Figure 3Cb*), as observed by confocal microscopy (*Figure 2B*). At the anastomosis point between ASVs and the primitive coronary plexus, a vessel continued to the aorta, forming a novel (secondary) orifice (*Figure 3Cc*; also see *Figure 2Bc*, *Figure 2—figure supplement 1B*). At 13.5 dpc, the orifice of the carotid artery was lost and the secondary orifice became larger (*Figure 3Cd*). In quail embryos, ASVs were more finely reticulated than in mouse embryos, making it difficult to track their pathways; nevertheless, we found endothelial invasion into the aortic wall from the ASV network at the aortic root at stage 30 (*Figure 3C*). These results indicate that the amniote ASVs are formed as a continuous endothelial structure from the pharyngeal region to the vascular network around the aortic root and contribute to the formation of the secondary orifice of the coronary artery. Thus, the unique coronary arteries developed stepwise in amniotes through remodeling, establishing novel orifices, and reorganizing the vascular networks.

## Amphibians retain ASV-like vascular networks throughout their lives

To understand the development of coronary vessels in amphibians after metamorphosis, we examined the anatomy of some frogs and newts. In breeding-aged *H. japonica*, the orifice of the outflow coronary vessels opened in the left branch of the truncus arteriosus, particularly at the root of the carotid trunk (*Figure 4A*). Thus, the topographical position of the coronary vessel was the same as at the juvenile stage (stage 46; *Figure 2C*). Histologically, these vessels were distributed on the surface of the outflow tract, but not on the ventricle. Examination of the ink-injected hearts of the American bullfrog (*Lithobates catesbeianus*) was consistent with this result (*Figure 4B*).

In the aquatic frog *Xenopus laevis* at stage 54, the outflow coronary vessels were first found at the point of the outflow tract bifurcation, particularly at the root of the left branch of the truncus arteriosus (*Figure 4C*). At stage 60, the arterial plexus expanded on the outflow tract to reach the aortic root. This vascular pattern was maintained throughout the development into the adult stage. On the other hand, no blood vessels were found on the ventricular surface. A histological section of the truncus arteriosus revealed some internal septa and the orifice of the outflow coronary vessels open at the root of the carotid trunk (*Figure 4Ca*).

We found the above vascular distribution to be conserved not only in Anura but also in Urodela, such as the breeding-aged Japanese fire-belly newt (*Cynops pyrrhogaster*). This specie also had ASV-like vessels above the outflow tract, whose orifice opened into the left branch of the truncus arteriosus (*Figure 4D*).

These findings suggest that frogs develop a vascular network that is morphologically similar to that of the ASVs and is arranged directly on the outflow tract as thick truncal coronary vessels, whereas the ventricle has no vascular network on its surface. Given the similarity of adult anatomical structures, the development of these ASV-like vessels is likely common to anuran and urodele amphibians.

## Coronary veins are conserved among tetrapods

As we found that coronary arteries are divergent between amniotes and amphibians, we examined coronary veins in this respect. In amniotes, blood supplied by coronary arteries enters coronary veins and is collected in one trunk, which opens into the sinus venosus (*Grant and Regnier, 1926*; *Figure 5A*). In 15.5 dpc mouse embryos, the connections of the coronary artery and the vein were established (*Figure 1—figure supplement 1*).

In amphibians, a vascular system distinct from truncal ASV-like vessels was found in the outflow tract, which connected to a single tube and opened into the sinus venosus (*Figure 5B and C*). Although peripheral vessels were not distributed on the ventricle, as in amniotes, their brancheing

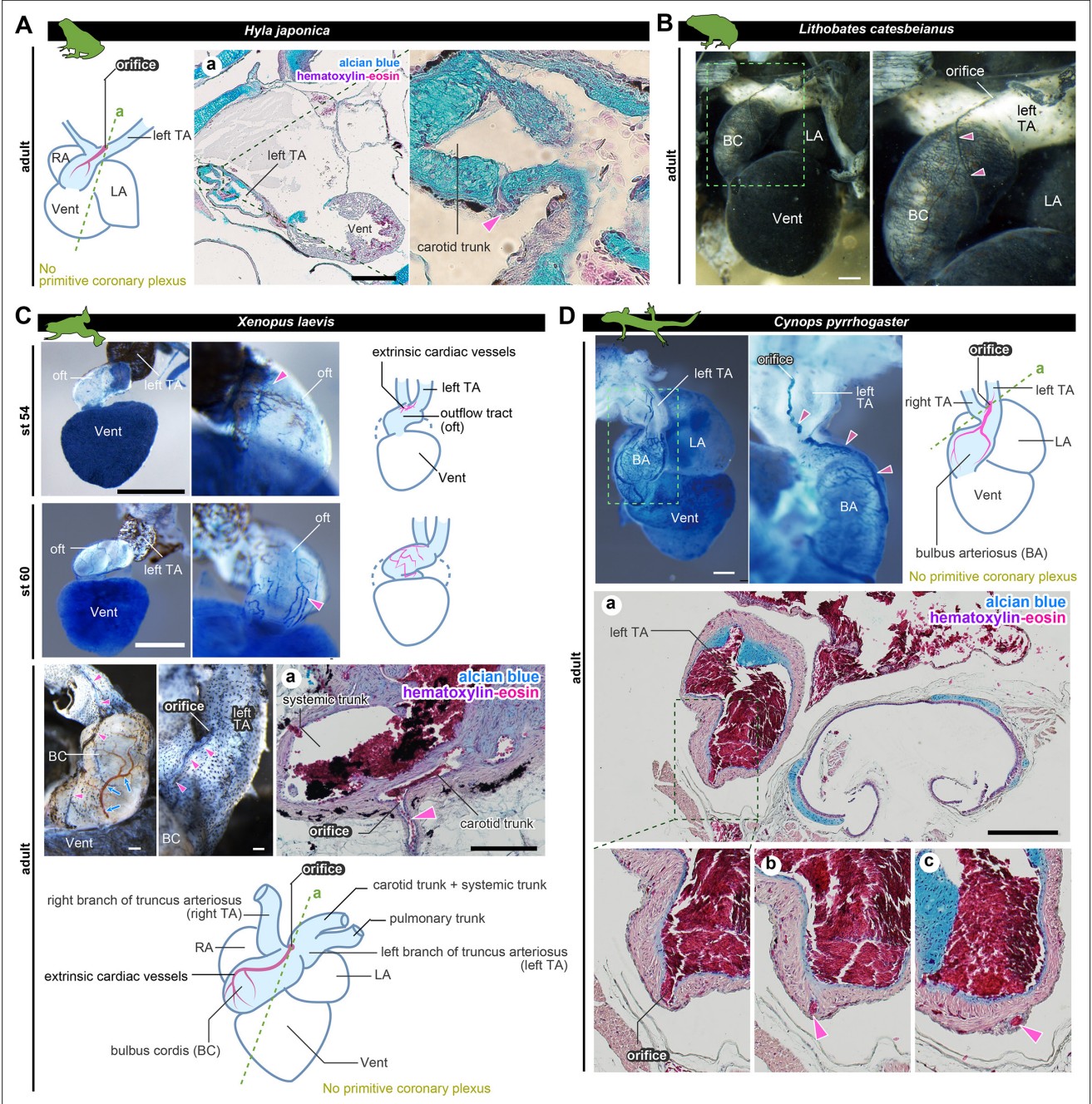

**Figure 4.** The positions of the orifices of the coronary arteries. (**A**) In the Japanese tree frog (*Hyla japonica*), the ASV-like vessels (pink arrowheads) branched from the aortic trunk close to the root of the carotid artery. (**B**) Ink-injected heart of African bullfrog (*Lithobates catesbeianus*; tadpole). (**C**) Development of the extrinsic cardiac arteries of *Xenopus laevis*. Ink injection visualized the blood vessels. In adult *X. laevis*, the ASV-like vessels branched from the carotid artery in the aortic trunk (**a**). (**D**) Ink-injected heart of the adult Japanese fire-belly newt (*Cynops pyrrhogaster*). The boxed area in the left image is enlarged and slightly rotated so that the distribution of blood vessels can be easily seen. Histological sections (a–c) show that the ASV-like vessels had an orifice at the root of the carotid artery (**a**), and it continued in the outflow tract (**b, c**). Arrowheads indicate extrinsic arteries on the outflow tract. Scale bars: 500 µm.

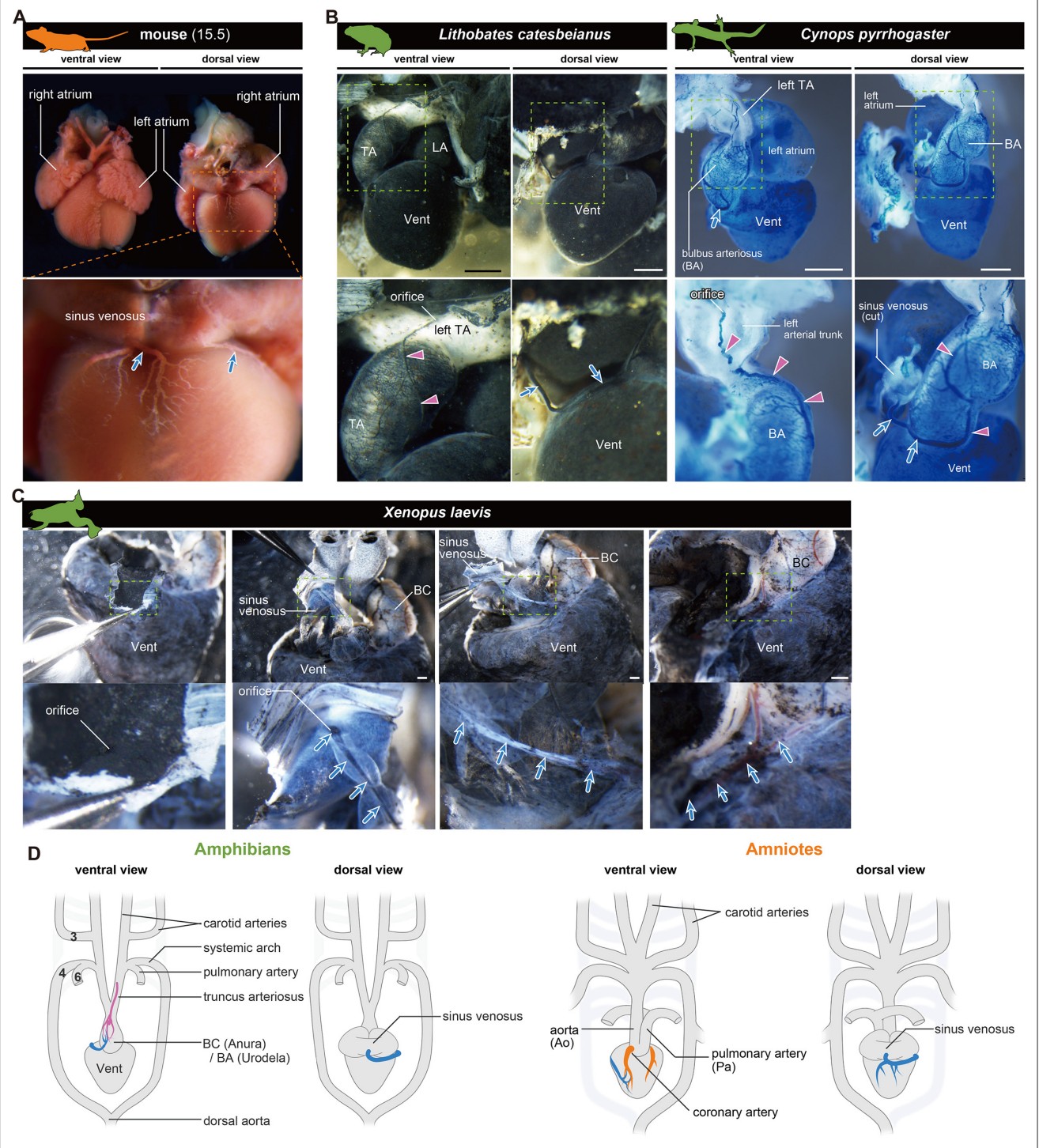

**Figure 5.** The orifices of the coronary veins are conserved among tetrapods. Ink-injected hearts are shown. (**A**) Mouse. (**B**) American bullfrog (*Lithobates catesbeianus*) and the Japanese fire-belly newt (*Cynops pyrrhogaster*). (**C**) *Xenopus laevis*. Arteries (pink arrowheads) and veins (blue arrows) were found on the surface of the outflow tract. In (**B**) and (**C**), the coronary vein (blue arrows) branched from the sinus venosus and reached the outflow tract. (**D**) Summary. BA, bulbus arteriosus; BC, bulbus cordis; TA, truncus arteriosus; Vent, ventricle Scale bars: 1 mm.

points were in the same position (*Figure 5D*), and this vessel should be regarded as a coronary vein in amphibians. Thus, unlike coronary arteries, the coronary vein is likely conserved between amniotes and amphibians.

## Coronary arteries in fishes

To further investigate the evolutionary history of coronary arteries, we checked their morphology in the zebrafish (*Danio rerio*) and chondrichthyans. In adult *flk1:egfp* zebrafish (72 days post-fertilization, dpf), a peripheral network of coronary vessels was found throughout the entire surface of the ventricle (*Figure 6A*). These ventricular coronary arteries led to the heart over a long distance directly from the hypobranchial artery, which branched off from the branchial arches (*Figure 6A and B*), in agreement with *Hu et al., 2001*. Before reaching the ventricle, this hypobranchial artery bifurcated and its two branches extended ventrally and dorsally to connect to the vascular network on the ventricular wall. Unlike the ASVs of amniotes and in sharp contrast to the ASV-like vessels of amphibians, these vessels in the zebrafish were not distributed in the walls of the outflow tract.

At the late pharyngula stage (31–38 hr post-fertilization, hpf) of *flk1:egfp* zebrafish embryos, the Y-shaped hypobranchial artery appeared in the midline of the pharyngeal region, and its lateral branches bilaterally connected with the first pharyngeal arteries of the mandibular arch (*Figure 6—figure supplement 1A*). The mid-caudal branch of the Y-shaped artery extended caudally toward the cardiac outflow tract. After hatching (48–72 hpf), it further extended along the ventral aorta (3–14 dpf) and formed a vascular plexus on the aorta around 21 dpf (*Figure 6—figure supplement 1A*). Around 36 dpf (juvenile stage), the hypobranchial artery was identified as a vessel running on the ventral side of the ventral aorta (*Figure 6B*). At 45 dpf, a vascular plexus was formed on the ventricle close to the aortic root, and part of it ttransiently covered the bulbus arteriosus (*Figure 6—figure supplement 1B*). At 46 dpf, it formed connections with the mid-caudal elongation of the hypobranchial artery (*Figure 6C*, *Figure 6—figure supplement 1B*). The formation of ventricular coronary arteries was complete by around 50 dpf, slightly before the adult stage.

In the zebrafish, the hypobranchial artery originates in the region of branchial arteries and the ventricular vascular plexus originates at the ventricle; in this respect, they are similar to ASVs and the primitive coronary plexus in amniotes, respectively. However, the zebrafish does not form a novel orifice in proximity to the heart and does not lose or reconstitute the above vessels. The hypobranchial artery is not a temporary structure and is maintained as the trunk of the coronary artery even in adults.

In chondrichthyans, we found anatomical features similar to those of adult zebrafish, although their arteries supply the wall of the outflow tract (*Figure 6D*). In *Lamna* sp. and *Narke japonica*, the coronary artery originated from a vessel further cephalad than the bulbus arteriosus (outflow tract). In *Deania calcea*, the arterial trunk branched from the branchial artery bifurcated on the conus arteriosus (outflow tract) and then supplied its peripheral branches to the ventricle.

## Histological structure of the ventricles

We also compared the histology of the ventricles in fetuses and adults of each species to gain insights into the relationships between the morphology of coronary vasculature and ventricular structures (*Figure 7*).

Mice and quails have thick ventricular walls. Even in 17.5 dpc murine fetuses, the myocardial wall was thick, albeit not as thick as in adult quails. On the other hand, the ventricular wall of amphibians was spongy and thin, even in adults. In the non-tetrapod sarcopterygians lungfish and coelacanth, ventricular walls were also spongy. Because of the limited resolution, we could not find coronary arteries in these two species. The previous studies suggest that the coronary arteries of lungfish are distributed exclusively in the outflow tract, like those of amphibians (*Szidon et al., 1969*; *Laurent et al., 1978*). However, zebrafish and sharks, which have ventricular coronary arteries, also have spongy ventricles. Although chondrichthyans have compact layers, they are also thin (*Icardo, 2017*). In adult *Lamna* sp., the lumen was fimbriated and reached close to the wall.

Thus, histologically, there does not seem to be a clear correlation between the presence or absence of coronary circulation in the ventricle and its histological structure. Our results suggest that the myocardium of the ventricles of mice and quails is thick and uniform rather than spongy. However, *Jensen et al., 2013a*; *Jensen et al., 2013b* suggested that lizard and snake hearts are spongy,

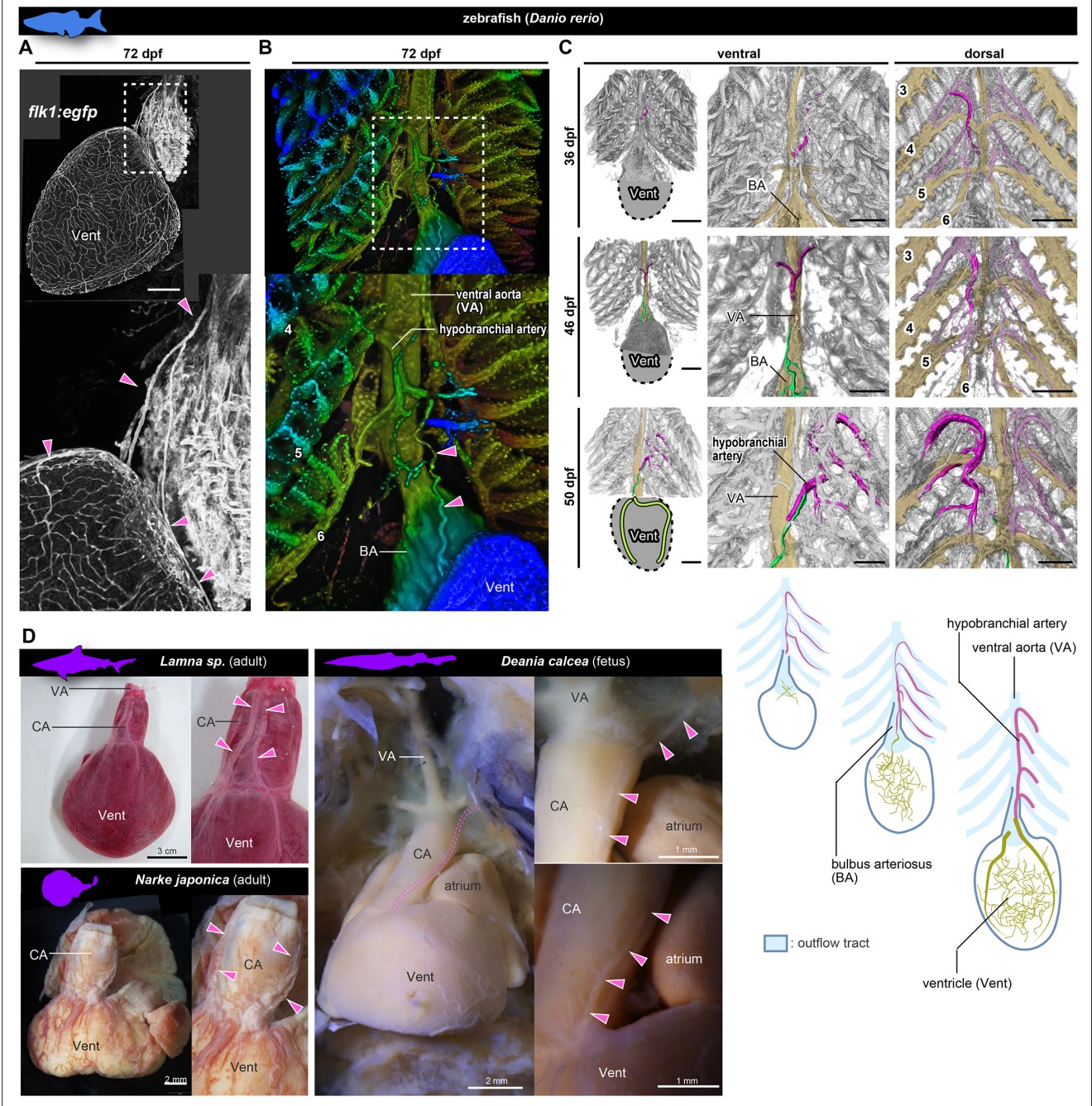

**Figure 6.** Anatomy and development of the coronary arteries of fishes. (**A, B**) *flk1:egfp* zebrafish at 72 dpf (juvenile). (**B**) Three-dimensional image. (**C**) *flk1:egfp* zebrafish at 36, 46, and 50 dpf. The hypobranchial arteries (pink) arose from the dorsal side of the pharyngeal arch arteries. The vascular network (green) appeared on the ventricular surface at 46 dpf and connected with the hypobranchial artery at 50 dpf. The numbers indicate the branchial arteries. (**D**) Chondrichthyans. The anatomical pattern of the hypobranchial and coronary arteries was identical to that of zebrafish. BA, bulbus arteriosus; CA, conus arteriosus; VA, ventral aorta; Vent, ventricle. Scale bars: 100 µm.

The online version of this article includes the following figure supplement(s) for figure 6:

**Figure supplement 1.** The development of hypobranchial artery in *flk1:egfp* zebrafish.

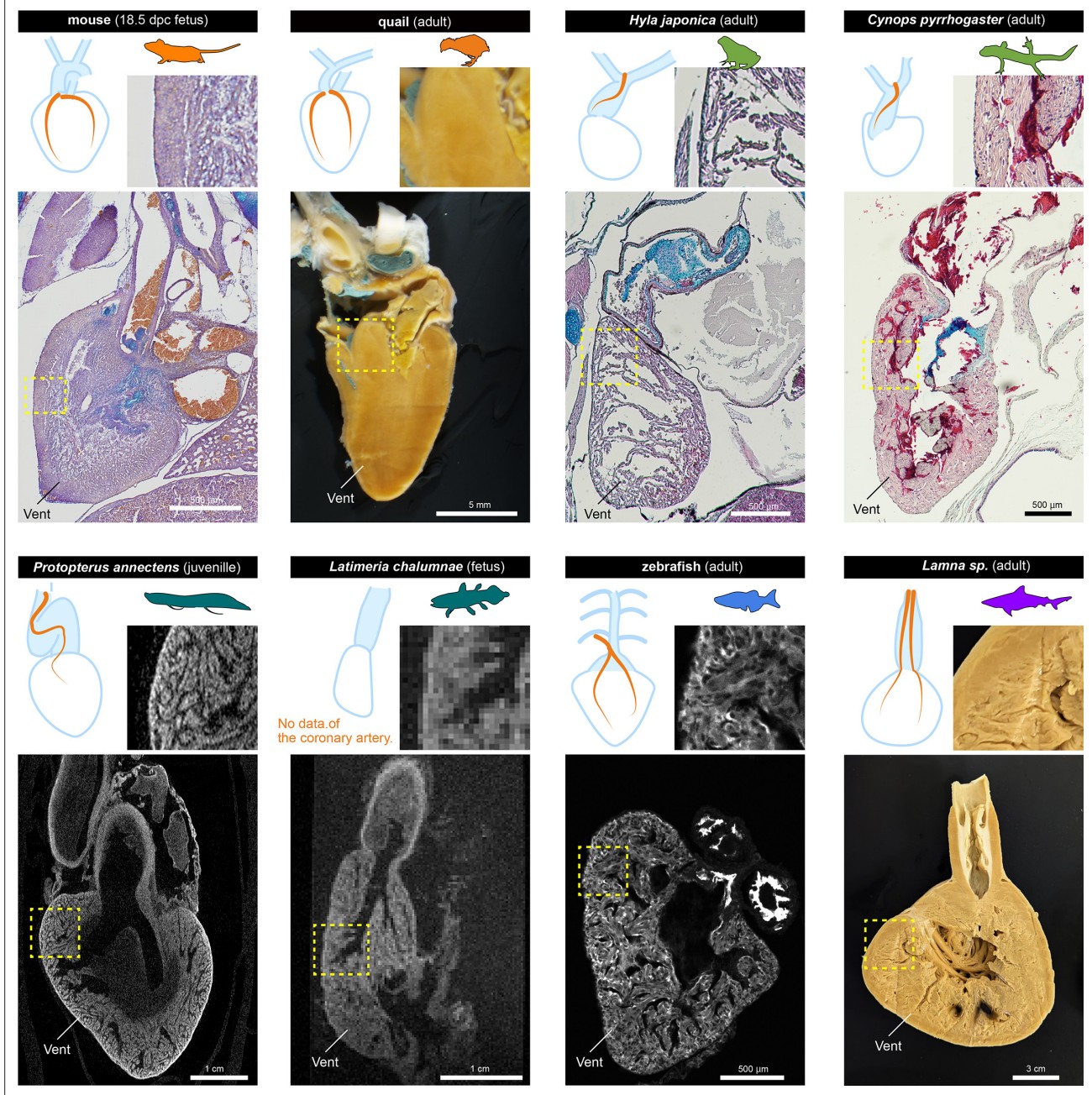

**Figure 7.** The histological structures of the ventricles. The sagittal sections of the hearts of various jawed vertebrates. For the mouse, *Hyla japonica*, and *Cynops pyrrhogaster* histological sections stained with HE and Alcian blue are shown. For the lungfish (*P. annectens*) and coelacanth (*L. chalumnae*), micro CT and MRI scanned images are shown, respectively. Quail and shark (*Lamna* sp.) hearts were cut by hand. The distribution of coronary arteries is shown in the schemes. We could not observe any coronary arteries in the lungfish and coelacanths because of low resolution; the distribution of the arteries in the lungfish is based on previous reports (*Szidon et al., 1969*; *Laurent et al., 1978*). Vent, ventricle.

whereas they have the amniote-type ventricular coronary arteries (*MacKinnon and Heatwole, 1981*), indicating that the establishment of these arteries may not correlate with ventricular wall thickness.

## Discussion

### ASVs as ancestral coronary arteries

Since Ibn al-Nafis discovered coronary arteries in the 13th century (*Numan, 2014*), their developmental origin has been controversial. Traditionally, it was believed that ventricular coronary arteries originate from the aortic wall in amniotes through outgrowth and sprouting angiogenesis (*Bennett, 1936*; *Hutchins et al., 1988*). However, more recent studies have demonstrated in mice that amniote-type ventricular coronary arteries develop through the reconstitution of embryonic vessels (ASVs) and the primitive coronary plexus, with the ingrowth formation of orifices (*Bogers et al., 1989*; *Red-Horse et al., 2010*; *Peng et al., 2013*; *Tian et al., 2013*; *Chen et al., 2014*; *Ivins et al., 2015*; *He and Zhou, 2018*). Our present results suggest that the coronary artery develops via reconstitution of the primitive vessels in mice and quails and this developmental process should be unique to amniotes.

In frogs and newts, we found that cardiac vessels are limited to the truncal outflow region and are composed of ASV-like vessels and the coronary vein. Frogs and newts have no primitive coronary plexus on the ventricle. Accordingly, remodeling to ventricular coronary arteries with secondary orifices does not occur in amphibians. These truncal ASV-like vessels of amphibians are identical to the ASVs of amniotes in topographical relationships: their orifices open at the root of the carotid artery (third pharyngeal artery derivative) and are distributed in the outflow tract in the same way. Given these anatomical connections, the ASVs of amniotes and ASV-like vessels of amphibians should be morphologically homologous.

The molecular identity of these vessels remains uncertain. In mice, ASVs are considered blood vessels because they contain erythrocytes and connect to the aortic endothelium; however, a subset of early ASV endothelial cells express *Prox1*, *Vegfr3*, and *Lyve1*, which are markers for lymphatic vessels (*Chen et al., 2014*; *Maruyama et al., 2019*). This suggests a lymphatic vessel–like identity of mouse ASVs, although they function as blood vessels, and their origin has been speculated to be from $Prox1^+$ cells, which also give rise to cardiac lymphatic vessels (*Maruyama et al., 2021*). The cardiac expression of *Prox1*, *Vegfr3*, and *Lyve1* orthologs in amphibians remains unknown. Recent identification of several endothelium-specific transcription factors by high-throughput single-cell RNA sequencing (*Liao et al., 2022*) will enable the comparisons of the amphibian vessels with ASV-specific molecular markers.

To determine whether the coronary circulation of anurans and urodeles is ancestral to that of amniotes, it is necessary to perform outgroup comparisons. Caecilians, the third group of extant amphibians, have a unique coronary circulation that is distinct from that of any other vertebrate group. In *Hypogeophis rostratus*, the so-called coronary arteries supply the heart tissue directly from the trabecular spaces of the ventricle (*Lawson, 1966*), and *Epicrionops* reportedly has no coronary arteries (*Wilkinson, 1996*). Thus, the morphology of caecilian arteries differs from that of ASV-like vessels. On the other hand, the coronary veins of caecilians branch from the sinus venosus (*Lawson, 1966*; *Wilkinson, 1996*), as observed in mice, frogs, and newts (*Figure 5*).

We suggest that caecilian coronary arteries are unlikely to be ancestral to those of other amphibians. Vessels similar to ASV-like vessels, which branch off the third pharyngeal arch artery, are distributed on the outflow tract, and end at the level of the outflow–ventricle junction, have been observed in the African lungfish (*Protopterus aethiopicus*; *Szidon et al., 1969*; *Laurent et al., 1978*). The South American lungfish (*Lepidosiren paradoxa*) has similar vessels on the outflow tract, although their cranial part connects to the branchial arteries via the hypobranchial artery (*Foxon, 1950*), similar to our findings in zebrafish and shark. Coronary arteries originating from the branchial region and crossing the subepicardium of the coronary outflow tract are shared by the actinopterygian bichir (*Polypterus senegalus*) and chondrichthyans, although the locations of their orifices are uncertain (*Durán et al., 2014*; *Lorenzale et al., 2018*). Thus, the vascular connection from the pharyngeal region to the outflow tract, as seen in ASVs, is phylogenetically ancestral, and coronary arteries unique to caecilians are a derived trait likely associated with their heart being located near the middle of the body, which is also a derived trait.

Histologically, ASV-like vessels in tadpoles are small and may have the identities of both lymphatic and blood vessels, similar to ASVs in mice. However, in adults, they thicken to form outflow tract vessels and contain numerous blood cells (*Figure 4*), indicating their function as mature blood vessels. Since these vessels constitute a network with coronary veins derived from the sinus venosus in their periphery, ASV-like vessels in tadpoles become arteries in adult amphibians. Comparison with the morphology of lungfish, bichir, and shark vessels suggests that amphibian ASV-like vessels are the ancestral form of coronary arterial vessels in jawed vertebrates, with ASVs in amniotes representing remnants of this form. While amniote ASVs have probably lost their arterial function, they may be maintained as embryonic primordia for the subsequent development. Thus, the existence of ASVs in amniotes may be explained as developmental burden (*Riedl, 1978*), which is typically exemplified by notochords in vertebrates (*Fujimoto et al., 2022*).

## Diversity of coronary arteries in fishes

Whereas the ancestral form of coronary arteries is conserved among mammals, birds, and amphibians (except caecilians) as ASVs or ASV-like vessels, the coronary circulation of fishes is highly diverse and whether it is homologous to that of tetrapods remains unclear (e.g. *Grant and Regnier, 1926*).

Coronary vasculature is most variable in teleosts. Fish hearts are often classified into four types-based on the histological distribution of coronary arteries on the ventricle (*Tota et al., 1983*; *Tota, 1989*; *Icardo, 2017*). Some teleosts lack any extrinsic cardiac circulation, but this has been regarded as a derived trait because non-teleost actinopterygians (e.g. bichirs) and chondrichthyans have cardiac vessels (*De Andrés et al., 1990*; *Farrell et al., 2012*; *Durán et al., 2014*; *Farrell and Smith, 2017*; *Lorenzale et al., 2018*). It is generally thought that the lower need for external oxygenation probably causes the lack of ventricular coronary circulation due to the acquisition of spongy heart walls. However, it is unclear whether and how the presence or absence of ventricular coronary arteries correlates with the histological characteristics of the ventricles (*Figure 7*). Among chondrichthyans, the posterior coronary arteries develop as branches of the subclavian arteries in some stingrays (*Muñoz-Chápuli et al., 1994*). Similar arteries are sometimes found even in amniotes, termed conus arteries, although they are distributed only on a small part of the ventricle (*Schlesinger et al., 1949*; *Simons, 1965*).

Creating a comprehensive scheme for the coronary system in fishes, especially teleosts, is challenging due to its complexity. However, on the basis of our observations of zebrafish, we can generally interpret teleost coronary arteries as originating from at least two vascular networks: a branchial region network and a primitive coronary plexus. Vessels from these networks anastomose at the boundary between the outflow tract and the ventricles, similar to anastomosis between ASVs and the primitive coronary plexus in amniotes. However, unlike in amniotes, they do not undergo remodeling and remain connected to form adult coronary arteries. Furthermore, these vessels are detached from the walls of the outflow tract and are connected to multiple branchial arteries through hypobranchial arteries, unlike ASV-like vessels in amphibians, bichir, and chondrichthyans (*Figure 6*). Because morphological comparison is difficult, we cannot identify the strict organ-level homology between ASVs and zebrafish hypobranchial arteries. Nevertheless, as the vascular structures derived from the pharyngeal arch to supply the aortic sinus, zebrafish hypobranchial arteries and related vessels undergo the same developmental processes in the same embryonic domain as ASVs, which should be considered as the same developmental module.

In zebrafish, *Paffett-Lugassy et al., 2017* reported that the hypobranchial artery endothelium is derived from the second heart field (termed cardiopharyngeal mesoderm by *Diogo et al., 2015*) in the pharyngeal arch. Although there is no direct evidence that amniote ASVs or amphibian ASV-like vessels originate from the second heart field or cardiopharyngeal mesoderm, the mouse periaortic lymphatic endothelium overlapping with ASVs originates from the *Islet1*-positive pharyngeal core mesoderm (*Maruyama et al., 2019*), and it is quite plausible that ASV distribution coincides with the mesodermal domains mentioned above. If the distribution of vascular structures corresponds to that of the mesenchyme, similar to the tight correspondence between distributions of neural crest cells and the trigeminal nerve in the face (*Higashiyama et al., 2021*), the variations in vascular distribution between species may indicate differences in the mesenchymal contribution to the heart.

It should be noted that adult ventricular coronary arteries differ in their endothelial cell lineage, even in the same class animals (Table 1 in *Kapuria et al., 2018*). For example, in zebrafish, ventricular

circulation is provided by the endocardium of the atrioventricular junction (*Harrison et al., 2015*); in gourami (*Trichogaster trichopterus*), it is derived solely from the hypobranchial arteries (*Shifatu et al., 2018*); in the giant danio (*Devario malabaricus*), it arises from the hypobranchial artery and the atrioventricular junction (*Shifatu et al., 2018*). These differences may be caused by the differences between species in the contribution of embryonic vessels. The developmental origin of the endothelial cells of the adult ventricular coronary vasculature also varies among amniotes (e.g. *Kapuria et al., 2018*), indicating that the relative contribution of ASVs and the primitive coronary plexus to adult coronary arteries varies among jawed vertebrates. It is not certain whether these cell lineage differences affect the vessels' morphological identity. If they do not affect it, it could be regarded as an example of developmental system drift (*True and Haag, 2001*), but further research is needed on this issue.

In summary, the origin of the coronary arteries of jawed vertebrates may be defined by the presence of the pharyngeal vessels (e.g., ASVs, amphibian ASV-like vessels, and hypobranchial arteries) and the primitive coronary plexus, and these two vascular networks have combined to produce a variety of coronary vasculature morphologies.

## Amniote evolution and the establishment of the true coronary artery

The causes of the acquisition of novel amniote-type coronary arteries through remodeling are unclear. Coronary artery remodeling in amniotes can hardly be attributed solely to functional changes in the ventricles because lizards and snakes have spongy ventricles (*Jensen et al., 2013a*; *Jensen et al., 2013b*) but also have the amniote-type coronary arteries. Thus, the amniote-specific coronary artery remodeling may be related to morphological changes in the pharyngeal region rather than to functional requirements (*Figure 8A*).

The cardiopharyngeal structures in tetrapods have progressively changed during the water-to-land transition to form the amniote bauplan adapted to arid land. Branchial arches have changed into neck structures, and a long neck has formed in the amniotes with the morpho-functional separation of the head and pectoral girdle. Simultaneously, the heart, located ventral to the pharynx, moved back into the thorax and became separated from the pharyngeal structures (*Hirasawa et al., 2016*). This heterotopic shift of the heart inevitably resulted in various reorganizations of the pharyngeal arch derivatives, such as the recurrent laryngeal nerves of the vagus, which have been modified into a peculiar route without altering their anatomical connections (*Higashiyama et al., 2016a*). It seems reasonable to speculate that such reorganization of the branchial arteries may have influenced the establishment of true coronary arteries with secondary orifices. Myocardial thickening in mammals and birds may have resulted from the pre-existence of amniote-type coronary arteries, making coronary arteries essential for survival.

Only one exception is known among amniotes. The orifice in turtles is in the middle of the carotid artery, as it is in frogs (*Spalteholz, 1924*; *Grant and Regnier, 1926*; *MacKinnon and Heatwole, 1981*), suggesting that the amniote-type coronary arteries were lost secondarily in the turtle lineage. Thus, transient ASV formation is conserved during development even after the acquisition of true coronary arteries, enabling them to return from the amniote type to the ancestral type. The loss of amniote-type coronary arteries in turtles might be related to the peculiar morphogenesis of the shoulder girdle in this lineage (*Nagashima et al., 2009*), but the extent to which this structural change affects cardiac function remains unknown.

The evolutionary origin of extrinsic cardiac arteries is still a mystery. Cyclostomes do not have coronary circulation, but whether this is the ancestral condition or a secondary loss is uncertain (*Grant and Regnier, 1926*). If they have retained the ancestral pattern, the establishment of extrinsic cardiac vessels may be related to the heterotopic shift of the cardiac region and head–trunk interface in the ancestor of gnathostomes (*Higashiyama et al., 2016a*), although further studies are warranted in this lineage.

## Clinical implications of the coronary artery development

Using the present theory, we can explain some congenital coronary artery abnormalities, an uncommon disease entity covering a broad spectrum of abnormalities (*Figure 8B*). Although many cases remain asymptomatic and are incidentally found by coronary angiography, these abnormalities sometimes cause sudden death (*Taylor et al., 1992*; *Yuan, 2014*; *Villa et al., 2016*). In particular, the anomalous

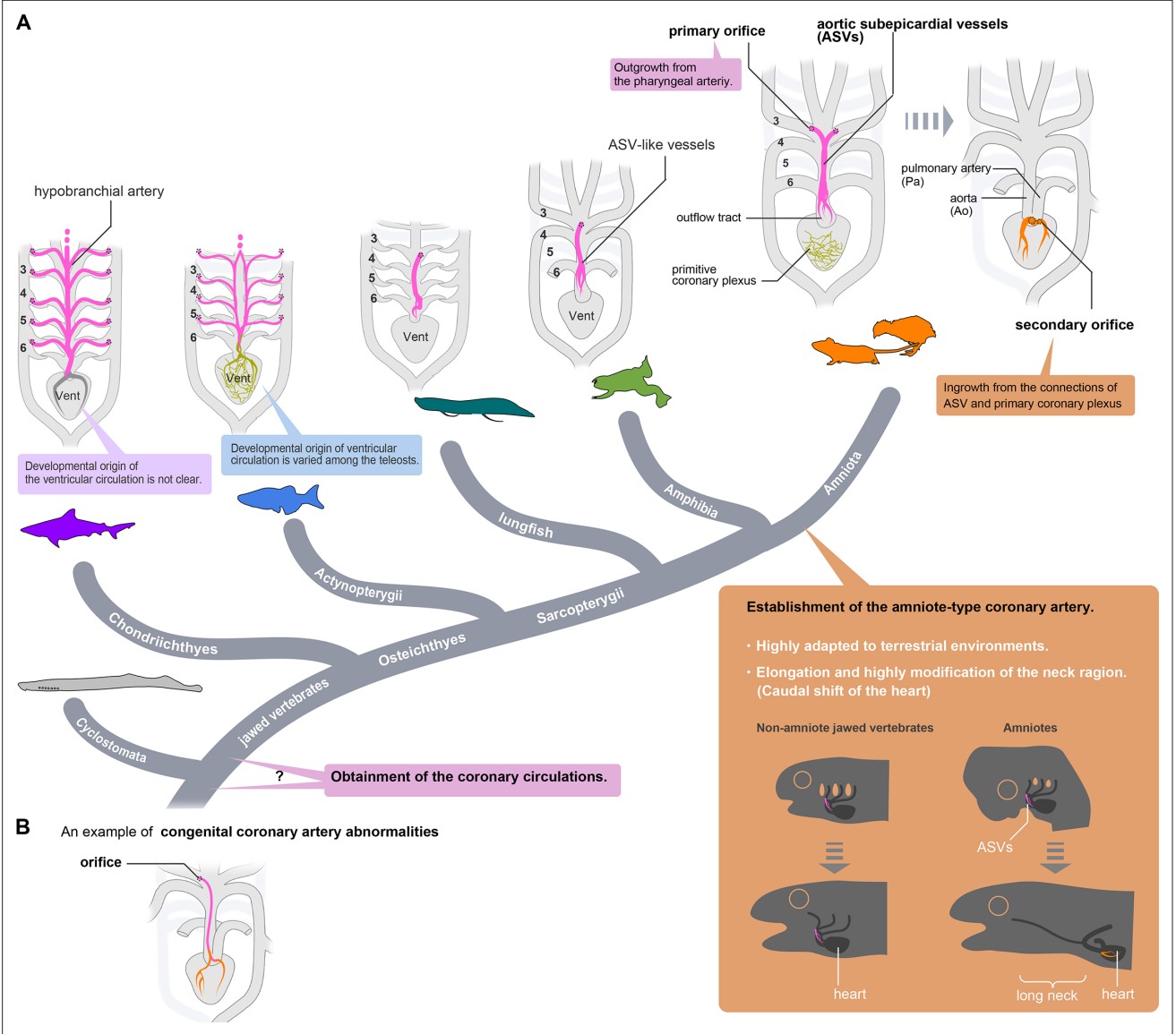

**Figure 8.** Evolution of coronary arteries. (**A**) The ventricular coronary arteries unique to amniotes are reconstituted from the ASVs and the primitive coronary plexus. The ASV-like vessels of amphibians and the hypobranchial arteries of fishes are comparable to the ASVs, rather than to the ventricular coronary arteries of amniotes. It remains unclear whether the ventricular artery in chondrichthyans arises as an extension of the hypobranchial artery or has a different developmental origin. As ventricular arteries have a variety of developmental sources in teleosts, ancestral vessels in this region may have had a variety of developmental origins in different animal species. In amniotes, such redundant coronary arterial development gave way to a single robust and unchangeable developmental program, resulting in developmental constraints. (**B**) A scheme of congenital coronary artery abnormalities according to *Kim et al., 2009*.

origin of the coronary artery from the innominate (brachiocephalic) artery or adjacent aortic arch is accompanied by other severe anomalies and is often lethal (*Davis and Lie, 1977*; *Asada et al., 2019*; *Pandey et al., 2019*), although some patients survive (*Santucci et al., 2001*; *Duran et al., 2008*; *Kim et al., 2009*). Since the brachiocephalic artery is derived from the right third pharyngeal arch artery and they pass across the ventral side of the ascending aorta, the anomalous coronary arteries in these reports are similar to ASVs. Thus, this type of anomaly can be considered persistent embryonic vasculature.

In conclusion, the present study suggests that the ventricular coronary artery, a common feature in amniotes, originated through ancestral cardiovascular remodeling during the evolution of amniotes. It is possible that this artery was necessary for the thickening of the ventricular wall, which is associated with high physiological activity. It is still undeniable that coronary artery formation through ASV

remodelling may be a convergent evolution of mammals and birds. The resemblance of coronary arterial morphology in adult squamates to that of mammals and birds may reflect a potential common developmental process, although further research is required to delve deeper into this aspect. This new understanding of the evolutionary history and morphology of coronary circulation will be useful in evolutionary research on amniotes and in comparisons among animal models to study coronary artery disease.

## Materials and methods

### Animals

Wild-type mice (*Mus musculus*) of the ICR background were kept in an environmentally controlled room at 23 ± 2°C with a relative humidity of 50–60% under a 12 hr light: 12 hr dark cycle. Embryonic ages were determined by timed mating, with the day of the plug being 0.5 dpc.

Eggs of Japanese quail (*Coturnix japonica*) were obtained from Motoki Hatchery (Saitama, Japan). They were incubated in a humidified atmosphere at 37 °C until the embryos reached appropriate stages (*Ainsworth et al., 2010*).

African clawed frogs (*Xenopus laevis*) were obtained from the Watanabe *Xenopus*-inbred strain resource center (Hyogo, Japan) and were kept at room temperature (RT). The embryonic stage was determined as in *Nieuwkoop et al., 1994*. Japanese tree frogs (*Hyla japonica*) were caught in Tsukuba (Ibaraki, Japan), and American bullfrogs (*Lithobates catesbeianus*) were caught in Oyamadairi Park (Hachioji, Tokyo, Japan) according to the laws of the Invasive Alien Species Act. Japanese fire-belly newts (*Cynops pyrrhogaster*) were obtained from Kingyozaka (Tokyo, Japan).

Zebrafish (*Danio rerio*) were maintained as described by *Fukuhara et al., 2014*. In addition to the wild-type zebrafish, we used Tg(*flk1:egfp*) mutant zebrafish according to *Fukuhara et al., 2014*. Embryos and larvae were staged by hpf and dpf at 28–28.5°C (*Kimmel et al., 1995*).

Chondrichthyan samples were obtained through fish markets, mainly from incidental bycatch. The hearts of salmon shark (*Lamna* sp.) were obtained from Yoshiike (Tokyo, Japan), Japanese sleeper rays (*Narke japonica*) were from Induka Shouten (Nagasaki, Japan), and the fetuses of bird-beak dogfish (*Deania calcea*) were from Gyosai-ya (Tokyo, Japan). Lungfish (*Protopterus annectens*) was purchased from Meitosuien (Nagoya, Japan) and the heart was dissected out under anesthesia with 0.2% tricaine methanesulfonate (MS-222, Sigma Chemical Co.). The heart of coelacanth (*Latimeria chalumnae*) was dissected from the juvenile individual used for other experiment (*Nikaido et al., 2013*).

Data for lungfish and coelacanth were obtained from a single sample, but all other data are based on at least four samples; as we found no differences among the samples, we did not specify the sample size in the text. All animal experiments were approved by the University of Tokyo Animal Care and Use Committee and were performed in accordance with the institutional guidelines and the Act on Welfare and Management of Animals.

### Histological sections

To prepare cryosections, samples were fixed in 4% paraformaldehyde (PFA) in PBS overnight at 4 °C and washed with PBS. The samples were transferred consecutively into 10% sucrose in PBS, 30% sucrose in PBS, and 30% sucrose in Optimal Cutting Temperature (OCT) compound (Tissue Tek, Sakura Finetek, Tokyo, Japan) for 30 min at 4 °C for each step, and embedded in 100% OCT compound. The embedded block was stored at −20 °C before use. The samples were cryosectioned (8–12 µm thick).

To prepare paraffin sections, samples were fixed with Serra's solution (a mixture of formalin, ethanol, and acetic acid at 7:2:1) overnight and stored in 70% ethanol at RT. The samples were dehydrated stepwise with an ethanol-to-xylene series and embedded in paraffin. The paraffin blocks were cut into 6- to 9-µm-thick sections. The sections were stained with Alcian blue, hematoxylin, and eosin by using the standard protocol (HE staining).

### Immunohistochemistry

Cryosections were blocked with 1% skim milk in PBS with 0.5% Tween-20 at RT for 15 min and incubated with primary monoclonal antibody (rat anti-mouse PECAM-1 [CD31; BD Pharmingen, San Diego, CA, USA; 1:400] or mouse monoclonal anti-QH-1 [Developmental Studies Hybridoma Bank; 1:100]) at 4 °C overnight. The sections were washed in PBS four times at RT and incubated with Alexa

Fluor–conjugated secondary antibodies (Abcam, 1:400) for 1 hr. The sections were washed in PBS five times and mounted with Aqua-Poly/Mount (Cosmo Bio Co., Ltd., Tokyo, Japan). Nuclei were visualized with DAPI (Molecular Probes, Eugene, OR, USA). Fluorescence signals were visualized under a computer-assisted confocal microscope (Nikon D-Eclipse C1), and images were processed using NIS Elements software (Nikon, Tokyo, Japan). The same protocol was followed for whole-mount hearts and embryos, except that they were incubated with secondary antibodies overnight. Some sections were incubated with biotin-conjugated secondary antibodies and visualized using the Vectastain ABC System (Vector Laboratories, Burlingame, CA, USA).

### Three-dimensional reconstruction

Images of histological sections or confocal images were loaded into the Amira software platform (Thermo Fisher Scientific, Waltham, MA, USA) with a voxel size appropriate to section thickness.

### Ink and latex injection

To visualize coronary arteries, *Xenopus laevis*, red-bellied newts, and American bullfrogs were anesthetized at appropriate stages on ice or with 0.1% tricaine methanesulfonate (MS-222; Wako, Japan). Ink (Kiwa-Guro, Sailor, Japan) was gently injected using a glass micropipette from the hepatic portal vein. The hearts were then collected and fixed in 4% PFA overnight at 4 °C. The samples were cleared by CUBIC solution (*Susaki et al., 2014*) and examined under a stereo microscope. Mouse and quail hearts were injected with latex or resin as in *Higashiyama et al., 2016b*.

### Visualization of the vascular system in zebrafish

Samples were fixed in 4% PFA overnight at 4 °C, trimmed, and immersed in the CUBIC solution overnight at RT. Fluorescence signals were visualized under the Nikon D-Eclipse C1 microscope; z-stack images (2–5 μm) were used.

### Live imaging of zebrafish

A multi-photon excitation microscope (FV1000MPE, Olympus, Tokyo, Japan) equipped with a water immersion 20×lens (XLUMPlanFL, 1.0NA, Olympus) was used. Sequential images were processed with FV10-ASW 3.1 viewer (Olympus) and analyzed with the Imaris software (Bitplane, Belfast, UK).

### Micro CT scan

The lungfish heart was fixed with 4% PFA and gradually dehydrated with the series of ethanol and then treated with 0.3% phosphotungstic acid (PTA; Wako, Japan) in 70% ethanol for 4 days according to *Metscher, 2009*. Images were acquired with a SkyScan 1172 scanner (Bruker) as following parameters; source voltage = 100 kV, source current = 100 μA, image pixel size = 10.14 μm, exposure = 295ms, and rotation step = 0.460 degree.

### MRI scan

The magnetic resonance image of the sample was acquired with a magnetic resonance imaging system using a vertical-bore 4.7 Tesla superconducting magnet (*Hashimoto et al., 2014*). The acquisition parameters are as follows: field of view (FOV)=25.6 mm cube, number of image matrix = 256 cube, voxel size = 100 microns cube, pulse sequence = three-dimensional spin-echo sequence, repetition time (TR)=200ms, echo time (TE)=12ms, number of signal accumulation = 2, and image acquisition time = 7.3 hr.

## Acknowledgements

We are grateful to Mayuko Kida and Akiyasu Iwase (The University of Tokyo) for helpful discussion. We are also grateful to Japanese fish markets, Yoshiike (Tokyo, Japan), Induka Shouten (Nagasaki, Japan), and Gyosai-ya (Tokyo, Japan), for providing the chondrichthyan samples and TOYO Corporation (Toyo, Japan) for assistance with micro CT analysis. We thank Naoko Yokota (The University of Tokyo) for technical assistance and Yuriko Kondo (The University of Tokyo) for her secretarial assistance. Fundings This work was supported by the Grants-in-Aid from the Japan Society for the Promotion of

Science (JSPS) 20H04858, 20K15858 (H.H.), JST, PRESTO JP22715256 (Y.A.), JSPS 22K07877 (S.M.T.), 19H01048, and 22H04991 (H.K.).

## Additional information

### Funding

| Funder | Grant reference number | Author |
|---|---|---|
| Japan Society for the Promotion of Science | 20H04858 | Hiroki Higashiyama |
| Japan Society for the Promotion of Science | 20K15858 | Hiroki Higashiyama |
| Japan Science and Technology Agency | PRESTO JP22715256 | Yuichiro Arima |
| Japan Society for the Promotion of Science | 22K07877 | Sachiko Miyagawa-Tomita |
| Japan Society for the Promotion of Science | 19H01048 | Hiroki Kurihara |
| Japan Society for the Promotion of Science | 22H04991 | Hiroki Kurihara |

The funders had no role in study design, data collection and interpretation, or the decision to submit the work for publication.

### Author contributions

Kaoru Mizukami, Conceptualization, Resources, Data curation, Software, Formal analysis, Supervision, Validation, Investigation, Methodology, Writing - original draft, Project administration, Writing – review and editing; Hiroki Higashiyama, Conceptualization, Resources, Data curation, Supervision, Funding acquisition, Validation, Investigation, Visualization, Methodology, Writing - original draft, Project administration, Writing – review and editing; Yuichiro Arima, Resources, Data curation, Validation, Investigation, Methodology, Writing – review and editing; Koji Ando, Sachiko Miyagawa-Tomita, Resources, Data curation, Writing – review and editing; Norihiro Okada, Resources; Katsumi Kose, Data curation, Methodology; Shigehito Yamada, Kazuko Koshiba-Takeuchi, Data curation; Jun K Takeuchi, Resources, Data curation; Shigetomo Fukuhara, Resources, Data curation, Methodology, Writing – review and editing; Hiroki Kurihara, Conceptualization, Resources, Supervision, Funding acquisition, Validation, Investigation, Project administration, Writing – review and editing

### Author ORCIDs

Hiroki Higashiyama (iD) http://orcid.org/0000-0003-1324-8139
Koji Ando (iD) http://orcid.org/0000-0002-4152-5706

### Ethics

All animal experiments were approved by the University of Tokyo Animal Care and Use Committee (approval ID: P19-043 and P19-050) and were performed in accordance with the institutional guidelines and the Act on Welfare and Management of Animals. We also follow the ARRIVE guidelines for animal research (Percie du Sert et al., 2020).

### Decision letter and Author response

Decision letter https://doi.org/10.7554/eLife.83005.sa1
Author response https://doi.org/10.7554/eLife.83005.sa2

## Additional files

### Supplementary files
• MDAR checklist

## Data availability

All data generated or analysed during this study are included in the manuscript and supplementary information. The genomic data of flk1:egfp zebrafish is available in ZFIN; Tg(kdrl:egfp)s843, ID: ZDB-ALT-050916-14 (https://zfin.org/action/feature/view/ZDB-ALT-050916-14).

The following previously published datasets were used:

| Author(s) | Year | Dataset title | Dataset URL | Database and Identifier |
|---|---|---|---|---|
| Stainier Lab | 2022 | s843Tg | https://zfin.org/action/feature/view/ZDB-ALT-050916-14 | ZFIN, ZDB-ALT-050916-14 |

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
