## [Editor Report]

Mizukami et al. investigate the evolutionary origins of coronary arteries in amniotes by comparing vascular morphologies across several species and developmental timepoints. They propose that coronary arteries are a novel structure in amniotes and that they arose from an ancestral vascular network surrounding the outflow tract. The ancestral network is similar to, and may be a remnant of, the transient aortic subepicardial vessels (ASVs) seen in developing mouse and quail hearts.

---

## [Decision Letter]

**Decision letter after peer review:**

Thank you for submitting your article "Coronary artery established through amniote evolution" for consideration by *eLife*. Your article has been reviewed by 3 peer reviewers, including Kristy Red-Horse as Reviewing Editor and Reviewer #1, and the evaluation has been overseen by Didier Stainier as the Senior Editor. The following individual involved in review of your submission has agreed to reveal their identity: Guo Huang (Reviewer #2).

Essential revisions:

1. Additional species are required to support conclusions about when different cardiac vessel configurations arose during evolution.

2. Some modifications to imaging are required to confirm conclusions on ASV attachment sites.

3. Please consider the physiology and functionality of the vessel beds presented, either with comments or new experiments.

4. Some clarity in the writing is necessary.

*Reviewer #1 (Recommendations for the authors):*

– Detailed descriptions of mouse (e.g., Chen, JCI, 2014) and zebrafish (e.g., Harrison, Dev Cell, 2015) vasculature and their development are already available making comparisons to new species the major novelty in the work presented and anatomical descriptions are somewhat shallow in terms of the information they provide about the vessels under study.

– The authors propose that the ASV/hypobrachial arteries are the ancestral coronary vessels. Given the shallow descriptions in the manuscript, the authors should provide additional information on the identity and function of these vessels and how these features are conserved.

– What endothelial subtypes are the ASVs in terms of molecular markers? Are they artery, vein, or capillary in molecular identity and is this conserved? Are they covered with smooth muscle (the commonly used smooth muscle marker works in many species, and likely would in the ones used here)? Chen et al., JCI, 2014 found molecular evidence that ASVs contained blood endothelial cells and lymphatic endothelial cells while Gancz et al. *eLife* found that there is a lymphatic plexus at this same location.

– Is the function of this vessel to provide oxygenation to the outflow tract or something else? Is there a capillary bed branching from the ASV arteries that could carry out oxygenation or is it just a larger vessel that might have a different function? Is this region of the outflow tract experiencing hypoxia despite the ASVs as defined by a chemical hypoxia probe? The authors should explore this issue.

– There is an insufficient selection of most groups of vertebrate diversity to justify many of the claims made in the manuscript. Mouse and quail alone are not a sufficient sampling for all of amniote diversity. This is particularly problematic in the fish sampling, who as correctly pointed out, have incredibly diverse coronary vasculature structures. A greater actinopterygin sample is needed and a sarcopterygian would be interesting, if one with coronary vessels is known.

– The developmental series in figure 2 is insufficiently sampled. Only one non-amniote is surveyed and an insufficient number of developmental timepoints sampled for *Xenopus*. Without more *Xenopus* sampling there could be a transient plexus that is lost between time points or one that forms later? Without a broader amphibian sampling we do not know that *Xenopus* is a proper representative of extant amphibians.

– The author's claims would be bolstered if they expanded their sample sized beyond N=2 so that the finding are considered robust statistically.

– The origination of the coronary orifice, in figure 3, is one of the most compelling part of the manuscript. However, it would be more so if a non-anuran amphibian, such as cynops pyrrhogaster was included to eliminate the possibility of an anuran specific trait.

– Assumptions about homology from structural caparisons could be greatly strengthened by lineage tracing experiments in the subset of organisms in which this is commonplace (zebrafish, mouse, avian).

*Reviewer #2 (Recommendations for the authors):*

1) The authors should include analysis or discussion of the coronary artery structure of a cold-blooded amniote species, such as lizard or snake, as the use of only mice and quail to represent amniotes makes it possible the evolutionary differences observed may be attributed to endothermic ability. Endothermic metabolism is strongly correlated with cardiac function and structure, so it may be reasonable to think that coronary arteries on the ventricle surface may have emerged to support the endothermic metabolism. If the structure of reptile arteries more closely resembles the non-amniotes, then the authors' claims of these morphological differences as developing between amniotes and non-amniotes would not be supported. If full morphological analyses of a reptile species cannot be performed, these limitations and existing literature on reptilian coronary artery structure should be discussed (https://doi.org/10.1111/brv.12056).

2) The authors should discuss the significance of the development of coronary arteries on the ventricle surface in terms of its physiological implications. The differences in vessel structure raises the question of why one configuration may be more advantageous over the other, so there should be some rationalization of why these changes have occurred throughout evolution and development. For example, it may be possible that arteries on the ventricle surface allow the heart to receive higher-oxygenated blood, allowing for greater cardiac function. Or perhaps the change of patterning arose to accommodate the denser and thicker ventricle walls seen in mammals and bird compared to non-amniotes. Such physiological implications should be included in the introduction and discussion to elevate the significance of the authors' observations.

3) The quality of images in the figures should be improved to improve clarity and strengthen the authors' claims, as their results are based on visual analysis of vasculature morphology. Specifically, Figures 1B, 2A-C, 3A-D and G, 4B-C, S1 and S2 struggle to show the intended evidence in a convincing manner and would greatly benefit from improved image quality.

*Reviewer #3 (Recommendations for the authors):*

Some additional comments that could help to improve the current manuscript for the authors to consider.

1. In discussion, "We found that amniote type coronary arteries developed by the reconstruction of primary vessels."

– What is the "primary vessel" in this context?

– Do the amniote type coronary arteries refer to ventricular coronary arteries? If so, the manuscript does not seem to describe a developmental origin of ventricular vessels.

2. Similarly, lines 307-308 "the present study shows that the true (amniote-type) coronary artery first developed by reconstitution of the ancestral cardiac vessels in amniote evolution". The word reconstruction suggests that amniote ventricular coronary arteries are derivatives of something exiting in an ancestral species. However, other parts of this discussion refer to amniote vessels as a novelty (line 260). These seemly conflicting statements could be modified for clarity.

3. The section Diversity of coronary arteries among vertebrates could be made more succinct and easier to read. For instance, discussion of the nematode vulva does not add to the section.

4. Is there another term besides "so called arteries" that the authors could use? This terminology reads rather casually for a scientific paper.

---

## [Author Response]

Reviewer #1 (Recommendations for the authors):– Detailed descriptions of mouse (e.g., Chen, JCI, 2014) and zebrafish (e.g., Harrison, Dev Cell, 2015) vasculature and their development are already available making comparisons to new species the major novelty in the work presented and anatomical descriptions are somewhat shallow in terms of the information they provide about the vessels under study.

We added three-dimensional reconstructions to compare newly acquired data on mice and frogs (Figure 2). Additionally, we included data on amphibians to support and clarify our arguments. Based on these findings, we have restructured the paper to highlight our contributions better. We believe that our study now presents a unique perspective on the anatomy and evolution of coronary arteries distinguishing it from previous descriptions of mice and zebrafish.

– The authors propose that the ASV/hypobrachial arteries are the ancestral coronary vessels. Given the shallow descriptions in the manuscript, the authors should provide additional information on the identity and function of these vessels and how these features are conserved.

The discussion of homology is a challenging task, and finding evidence to support it is even more difficult. It would be ideal if molecular markers could be identified and compared to determine commonalities between blood vessels, as the authors suggest. However, finding such markers for site-specific structures located far apart in distant lineages is uncommon. For instance, the middle ear bones of mammals and the jaw joint bones of chickens share the same genes involved in chondro- and osteogenesis but do not share any specific molecules. Still, they are obviously homologous in the topographical relationships. The changes in vascular endothelium are so extensive that even mouse and quail require different antibodies, and the degree of molecular similarity does not always lead to a determination of homology. The relationship between molecular features and homology has been the subject of numerous debates, and the concept of Character Identity Mechanism (ChIM; DiFrisco et al., 2020) has recently been established. Thus, we believe that topographical relationships still provide the best basis for determining morphological homologies for specific vessels in the body plan.

We believe that the morphological homology between the present ASVs and the truncal coronary arteries found in the amphibian outflow tract is reinforced by the additional data. Of course, recent developments such as single-cell RNA-seq hold promise for providing new insights into molecular homology. Thus, we added functional considerations to the discussion around lines 258-267.

– What endothelial subtypes are the ASVs in terms of molecular markers? Are they artery, vein, or capillary in molecular identity and is this conserved? Are they covered with smooth muscle (the commonly used smooth muscle marker works in many species, and likely would in the ones used here)? Chen et al., JCI, 2014 found molecular evidence that ASVs contained blood endothelial cells and lymphatic endothelial cells while Gancz et al. eLife found that there is a lymphatic plexus at this same location.

This remains uncertain even in mice. ASVs are considered blood vessels as they contain erythrocytes and connect to the aortic endothelium; however, a subset of early ASV endothelial cells express Prox1, Vegfr3, and Lyve1, which are markers for lymphatic vessels (Chen et al., 2014; Maruyama et al., 2019). This suggests that ASVs in mice may have lymphatic vessel-like identity, although they are blood vessels, and the origin of ASVs in mice has been speculated to be from Prox1+ cells, which also give rise to cardiac lymphatic vessels (Maruyama et al., 2021).

We tested to the immunohistochemistry stain amphibian tissues with smooth muscle actin and some endothelial markers for comparison, but unfortunately none of them showed satisfactory staining not only in the ASV-like vasculature but also in any vasculature throughout the body. Actually, previous studies on endothelial immunohistochemistry in amphibians are rare, and it is possible that the systemic vascular endothelium as well as the ASV may differ somewhat between amniotes and amphibians.

Thus, it is difficult to accurately identify whether the ASVs in the present study are arterial, venous, or near lymphatic. To address this, the present study treated ASVs and ASV-like vessels in amphibians simply as "vessels" instead of arteries, as it was difficult to determine their precise identity.

However, since ASV-like vessels in adult amphibians are thicker and have blood cells in the lumen, as well as forming cardiac veins and plexus similar to those in amniotes, it is likely that they are arteries.

The above discussion was added in the manuscript (around lines 258-267 and 290-295).

– Is the function of this vessel to provide oxygenation to the outflow tract or something else? Is there a capillary bed branching from the ASV arteries that could carry out oxygenation or is it just a larger vessel that might have a different function? Is this region of the outflow tract experiencing hypoxia despite the ASVs as defined by a chemical hypoxia probe? The authors should explore this issue.

Although experiments could not be performed, the fact that ASVs in mice and quail and early ASV-like vessels in amphibians are only a few µm in diameter and not of a size that blood cells can easily pass through, suggests that in these animal embryos ASVs probably have a limited function in oxygen supply and mainly make secondary orifice We expect that they act as signaling centers. However, in adult amphibians they would clearly have an oxygen-supplying role, as there are many blood cells inside. It is just as if the notochord, which functioned as the ancestral axial skeleton, transiently emerged as a signaling center during the vertebrate development.

We added this description in lines 290-301.

– There is an insufficient selection of most groups of vertebrate diversity to justify many of the claims made in the manuscript. Mouse and quail alone are not a sufficient sampling for all of amniote diversity. This is particularly problematic in the fish sampling, who as correctly pointed out, have incredibly diverse coronary vasculature structures. A greater actinopterygin sample is needed and a sarcopterygian would be interesting, if one with coronary vessels is known.

Due to limited access to embryos of multiple amniote species, we expanded our investigation to include the literature on adult coronary arteries in reptiles, such as lizards (Spalteholz, 1924; Erhart, 1935; MacKinnon and Heatwole, 1981; Farrell et al., 2012; also see Figure 1A). We also conducted the observations of the development of *Hyla japonica* and the fetuses of non-tetrapod sarcopterygians, such as lungfish and coelacanth. Although we were unable to identify coronary arteries in lungfish and coelacanth, previous literature on lungfish supports the idea that the distribution of coronary arteries in amphibians is highly consistent with that of lungfish (Szidon et al., 1969; Laurent et al., 1978). Thus, we can infer that the amphibian pattern is ancestral among phylogenetically sarcopterygians.

While actinopterygians exhibit a great deal of diversity, we were unable to obtain additional data. As a result, we have tempered our assertions regarding the homology of hypobranchial arteries in actinopterygians in the present paper. Nevertheless, information from lungfish and chondriichthyans suggests that the peculiar coronary arteries observed in Actinopterygians are largely phylogenetically derived modifications (see Discussion part).

– The developmental series in figure 2 is insufficiently sampled. Only one non-amniote is surveyed and an insufficient number of developmental timepoints sampled for *Xenopus*. Without more *Xenopus* sampling there could be a transient plexus that is lost between time points or one that forms later? Without a broader amphibian sampling we do not know that *Xenopus* is a proper representative of extant amphibians.

We newly obtained the three-dimensional imaging data to compare the developmental process of *H. japonica* with that of mice (Figure 2). We chose *H. japonica* because it undergoes metamorphosis from an underwater tadpole to a fully terrestrial adult, which is a more dynamic process than that of *Xenopus*, which remains aquatic throughout its life. The results showed that the ASV-like vessels in H. japonica stayed in the same area from the beginning of development, which is consistent with the development of *Xenopus* and with the anatomy other amphibians (Figure 4). We also added a histological section of the adult urodele, *Cynops pyrrhogaster* (Figure 4).

All these data indicate that in amphibians, ASV-like vessels are kept in the same position throughout development, with little difference in adult position between species.

– The author's claims would be bolstered if they expanded their sample sized beyond N=2 so that the finding are considered robust statistically.

Throughout the paper, the sample size was n=4 or more. The data for lungfish and coelacanth is only n=1, although the paper's claims are not problematic, as it supplements its discussion with findings from the previous research literature.

– The origination of the coronary orifice, in figure 3, is one of the most compelling part of the manuscript. However, it would be more so if a non-anuran amphibian, such as cynops pyrrhogaster was included to eliminate the possibility of an anuran specific trait.

We increased our results and discussion with a focus on the location of the coronary orifice. Unfortunately, we could not obtain the urodele embryos, but we added the data on adult orifice of Cynops pyrrhogaster (Figure 4). It was topographically same as that of anurans, suggesting that it is common in amphibians.

– Assumptions about homology from structural caparisons could be greatly strengthened by lineage tracing experiments in the subset of organisms in which this is commonplace (zebrafish, mouse, avian).

This experiment is difficult, and we could not add them in the present version. Also, we did not think that the lineage tracing is effective for the present study. This is because, whether ASVs or hypobranchial arteries, it is obvious that they originate from pharyngeal mesoderm because of their topographical positions. It might be effective to understand which particular pharyngeal arches produce them. But, in any case, this experiment is extremely difficult in amphibians.

Reviewer #2 (Recommendations for the authors):1) The authors should include analysis or discussion of the coronary artery structure of a cold-blooded amniote species, such as lizard or snake, as the use of only mice and quail to represent amniotes makes it possible the evolutionary differences observed may be attributed to endothermic ability. Endothermic metabolism is strongly correlated with cardiac function and structure, so it may be reasonable to think that coronary arteries on the ventricle surface may have emerged to support the endothermic metabolism. If the structure of reptile arteries more closely resembles the non-amniotes, then the authors' claims of these morphological differences as developing between amniotes and non-amniotes would not be supported. If full morphological analyses of a reptile species cannot be performed, these limitations and existing literature on reptilian coronary artery structure should be discussed (https://doi.org/10.1111/brv.12056).

We searched for non-mammalian and non-avian amniotic embryos, but this time we were unable to obtain them. We therefore referred to previous literature on reptile coronary arteries and hearts (e.g., Spalteholz, 1924; Erhart, 1935; MacKinnon and Heatwole, 1981; Farrell et al., 2012; Jensen et al., 2013a,b). The results showed that the anatomical connections of adult coronary arteries in lizards and snakes are conservative with those of mice and quails. A typical example is described in Figure 1A. We have also increased the description and discussion of reptile hearts throughout the paper.

2) The authors should discuss the significance of the development of coronary arteries on the ventricle surface in terms of its physiological implications. The differences in vessel structure raises the question of why one configuration may be more advantageous over the other, so there should be some rationalization of why these changes have occurred throughout evolution and development. For example, it may be possible that arteries on the ventricle surface allow the heart to receive higher-oxygenated blood, allowing for greater cardiac function. Or perhaps the change of patterning arose to accommodate the denser and thicker ventricle walls seen in mammals and bird compared to non-amniotes. Such physiological implications should be included in the introduction and discussion to elevate the significance of the authors' observations.

For this comment, we newly examined the correspondence between the distribution of coronary arteries and the structure of heart tissue and summarized it in Figure 7. The results suggest that the acquisition of amniote-type coronary arteries should not coincide with the evolution of changes in the structure of the ventricle. In other words, the evolution of coronary arteries itself was probably not caused by physiological demands. Perhaps it depends more on morphological changes in the pharyngeal arch artery and neck. However, the establishment of the amniote-type coronary artery may have enabled the development of thick myocardium in mammals and birds. We added this argument in the Results and the Discussion parts.

3) The quality of images in the figures should be improved to improve clarity and strengthen the authors' claims, as their results are based on visual analysis of vasculature morphology. Specifically, Figures 1B, 2A-C, 3A-D and G, 4B-C, S1 and S2 struggle to show the intended evidence in a convincing manner and would greatly benefit from improved image quality.

We improved all photos and layouts of all figures as possible.

Reviewer #3 (Recommendations for the authors):Some additional comments that could help to improve the current manuscript for the authors to consider.1. In discussion, "We found that amniote type coronary arteries developed by the reconstruction of primary vessels."– What is the "primary vessel" in this context?– Do the amniote type coronary arteries refer to ventricular coronary arteries? If so, the manuscript does not seem to describe a developmental origin of ventricular vessels.

This sentence was indeed inaccurate. We have completely rewritten the relevant section.

2. Similarly, lines 307-308 "the present study shows that the true (amniote-type) coronary artery first developed by reconstitution of the ancestral cardiac vessels in amniote evolution". The word reconstruction suggests that amniote ventricular coronary arteries are derivatives of something exiting in an ancestral species. However, other parts of this discussion refer to amniote vessels as a novelty (line 260). These seemly conflicting statements could be modified for clarity.

We have avoided the conclusion that the amniote-type ventricular coronary arteries are an evolutionary novelty. Throughout the paper, we emphasize that they are the result of remodeling during development.

3. The section Diversity of coronary arteries among vertebrates could be made more succinct and easier to read. For instance, discussion of the nematode vulva does not add to the section.

We reorganized and entirely rewrote the chapter. We decided to avoid the famous specific example of nematode vulva and cite True and Haag, 2001, which were the first to define DSD.

4. Is there another term besides "so called arteries" that the authors could use? This terminology reads rather casually for a scientific paper.

We rewrote to avoid confusion, using names such as "truncal coronary arteries" and "amniote-type ventricular coronary arteries" depending on the context. Only the strange coronary arteries of the caecilians, which we did not know the proper name for, remained "so-called coronary arteries."